# A Multi-Matrix Metabolomic Approach in Ringed Seals and Beluga Whales to Evaluate Contaminant and Climate-Related Stressors

**DOI:** 10.3390/metabo12090813

**Published:** 2022-08-30

**Authors:** Antoine É. Simond, Marie Noël, Lisa Loseto, Magali Houde, Jane Kirk, Ashley Elliott, Tanya M. Brown

**Affiliations:** 1Pacific Science Enterprise Centre, Fisheries and Oceans Canada, 4160 Marine Drive, West Vancouver, BC V7V 1N6, Canada; 2School of Resource and Environmental Management, Simon Fraser University, 4160 Marine Drive, West Vancouver, BC V7V 1N6, Canada; 3Ocean Wise, 101-440 Cambie Street, Vancouver, BC V6B 2N5, Canada; 4Freshwater Institute, Fisheries and Oceans Canada, 501 University Crescent, Winnipeg, MB R3T 2N6, Canada; 5Centre for Earth Observation Science, University of Manitoba, Winnipeg, MB R3T 2N2, Canada; 6Centre St-Laurent, Environment and Climate Change Canada, 105 McGill Street, Montreal, QC H2Y 2E7, Canada; 7Canada Centre for Inland Waters, Environment and Climate Change Canada, 867 Lakeshore Road, Burlington, ON L7S 1A1, Canada

**Keywords:** metabolites, ringed seals, beluga whales, mercury, organohalogens, climate change

## Abstract

As a high trophic-level species, ringed seals (*Pusa hispida*) and beluga whales (*Delphinapterus leucas*) are particularly vulnerable to elevated concentrations of biomagnifying contaminants, such as polychlorinated biphenyls (PCBs), polybrominated diphenyl ethers (PBDEs) and mercury (Hg). These species also face climate-change-related impacts which are leading to alterations in their diet and associated contaminant exposure. The metabolomic profile of marine mammal tissues and how it changes to environmental stressors is poorly understood. This study characterizes the profiles of 235 metabolites across plasma, liver, and inner and outer blubber in adult ringed seals and beluga whales and assesses how these profiles change as a consequence of contaminants and dietary changes. In both species, inner and outer blubber were characterized by a greater proportion of lipid classes, whereas the dominant metabolites in liver and plasma were amino acids, carbohydrates, biogenic amines and lysophosphatidylcholines. Several metabolite profiles in ringed seal plasma correlated with δ^13^C, while metabolite profiles in blubber were affected by hexabromobenzene in ringed seals and PBDEs and Hg in belugas. This study provides insight into inter-matrix similarities and differences across tissues and suggests that plasma and liver are more suitable for studying changes in diet, whereas liver and blubber are more suitable for studying the impacts of contaminants.

## 1. Introduction

High trophic-level marine mammals are particularly vulnerable to the effects of biomagnifying contaminants, including mercury (Hg) and persistent organic pollutants (POPs), such as polychlorinated biphenyls (PCBs) and polybrominated diphenyl ethers (PBDEs) [1,2,3,4,5,6,7]. Marine mammals with elevated concentrations of PCBs and PBDEs may experience perturbations of the regulation of the thyroid and estrogen axes, lipid metabolism, impaired reproduction, bone lesions and/or reduced immune function [4,8,9,10,11], whereas chronic Hg exposure may primarily affect their central nervous system, causing renal complications, hepatotoxicity, as well as immune and endocrine disfunction [6,12,13]. Diet is the main uptake pathway for contaminants in marine mammals. Due to their lipophilic nature, another significant route of contamination of POPs is the transfer from adult females to their young through the placenta and lactation [14,15,16,17]. Studies have shown that Hg can also undergo a relatively significant mother-to-offspring transfer in pinnipeds [18,19], but minimal transfer in cetaceans [20,21].

Arctic marine mammals are exposed to POPs and Hg as a result of the contamination of food webs from long-range atmospheric transport, as well as some local sources [3,22,23,24,25,26]. Although Arctic marine mammals tend to accumulate lower levels of these contaminants relative to populations living near urbanized areas in lower latitudes, concentrations in several Arctic ringed seal (*Pusa hispida*) and beluga whale (*Delphinapterus leucas*) populations have been reported to exceed marine mammal health effects thresholds [8,27,28], and adverse health effects have been reported for both species. For example, several liver detoxification-related gene transcripts, as well as hepatic and circulatory vitamin A, were related to blubber PCB concentrations in beluga whales from the western Canadian Arctic [29,30]. In the ringed seal population of Saglek Bay (Northern Labrador, Canada), Brown et al. [8] reported correlations between PCB blubber concentrations and liver transcript levels of genes involved in the detoxification of xenobiotics, the regulation of estrogens, glucocorticoids and cellular differentiation and proliferation.

Climate change presents another significant stressor for Arctic marine mammals. Changing sea-ice coverage and thickness, water chemistry, air and water temperatures, and snow and rain precipitation rates can lead to significant modifications in food web structures and dynamics, with implications for marine mammals and the fate of contaminants in the region [31,32,33,34,35]. Ringed seals are particularly vulnerable to such changes, as they depend on ice for foraging, breeding, resting, molting and protection against predators [36,37]. Although Canadian ringed seal populations are generally considered stable, their status in Canada has been designated of special concern, and a decline over the next three generations has been predicted due to changes in sea ice and snow cover in a rapidly warming Arctic [38]. Major declines in abundance and pup production in ringed seals from Hudson Bay (Canada) were associated with the changes of sea-ice conditions caused by climate change [39]. Reductions in body condition indices and ovulation rates (of up to 50%) in female ringed seals from Prince Albert Sound (Northwest Territories) have also been correlated with low ice years [40]. Climate-related changes may also affect the condition and health of other marine mammals, such as belugas. For example, changes in the physiology (i.e., decline in size at age), diet and Hg concentrations have been observed in Eastern Beaufort Sea (EBS) belugas, which may be the result of a reduction in the quality and/or availability of their preferred prey and, consequently, an increase in their energetic foraging expenditures [41,42,43,44].

Ringed seals and beluga whales are the most abundant circumpolar Arctic pinnipeds and odontocete species, respectively [38,42], as well as the most extensively monitored marine mammal species in the Canadian Arctic. Consequently, there have been many data collected on contaminants (e.g., Hg, PCBs, PBDEs) and ecological tracers (e.g., stable isotopes, fatty acid profiles, body mass/condition and prey mortality rates) over recent decades. Although there is growing evidence that marine mammal diets and contaminant exposure are influenced by a changing climate in Arctic ecosystems, the assessment and prediction of the individual and combined impact of these stresses on marine mammal populations remains understudied. Further, the most appropriate health assessment tools and tissues with which to characterize and monitor these impacts over time are yet to be determined.

Omics technologies (i.e., genomics, transcriptomics, proteomics, lipidomics and metabolomics) offer promising methods to assess contaminant and climate-change-induced impacts on marine mammals, having been applied successfully to identify transcriptomic and physiological responses in several fish and mammal species [4,8,30,45,46,47,48,49]. Metabolomics represents a promising tool for studying organism–environment interactions and for assessing organism function and health at the molecular level [45]. This technology aims to determine the profiles of a large suite of low-molecular-weight (<1000 Da) metabolites (e.g., amino acids, fatty acids, amines and sugars), and are the result of the downstream process of genomics, transcriptomics and proteomics combined [50]. Being at a higher level of biological organization, metabolites make it possible to assess an individual’s biological and/or functional response to environmental stressors.

Only a few studies have applied metabolomics to assess the impacts of such stressors on marine mammals. For example, Morris et al. [51] found several correlations between phosphatidylcholine hepatic profiles and blubber concentrations of several PCB and PBDE congeners in Canadian Arctic polar bear (*Ursus maritimus*) populations, and Simond et al. [4] reported correlations between short-chained chlorinated paraffins blubber concentrations and skin profiles of several fatty acids in St. Lawrence Estuary beluga whales. However, it was unclear whether those correlations reflected a disruption of lipid metabolism by xenobiotics or whether there were additional variables, such as diet, that may have contributed to these alterations.

The objectives of the present study were to (1) compare the metabolite profiles of multiple tissues (i.e., inner blubber, outer blubber, liver and plasma) in two marine mammal species (Labrador ringed seals and Eastern Beaufort Sea beluga whales) which are commonly used as indicator species across the Arctic, and (2) assess the impact of contaminants (e.g., PCBs, PBDEs, Hg) and the influence of the biological parameters most likely to be influenced by climate change on the health of these two species using a multi-matrix targeted metabolomics approach. This approach aims to provide insight into inter-matrix similarities and differences and to assist in the selection and validation of the optimal matrix for use in assessing the impacts of contaminants and/or climate-related stressors on the health of ringed seals and belugas.

## 2. Results

### 2.1. Biological Variables

Age, length, axial girth, blubber thickness, liver and muscle carbon and nitrogen-stable isotope ratios were determined for 10 adult male ringed seals harvested by Inuit partners from Lake Melville (NL, Canada), aged 6 to 26 years (Table 1). One ringed seal (#40) was not included in statistical analyses as its biological parameters (i.e., length, girth and stable isotope ratios) differed significantly from other individuals. This outlier was later identified as a *Put-en jar*, a category of ringed seals that spends more time offshore and has a predominantly oceanic diet compared to those ringed seals that are generally more coastal (Appendix A). Several biological variables were correlated. Length was positively correlated with axial girth (*r_s_* = 0.78; *p* = 0.01) and blubber thickness (*r^2^* = 0.86; *p* < 0.01). Stable isotope ratios δ^13^C were positively correlated with δ^15^N in muscle tissue (*r^2^* = 0.79; *p* = 0.01) (Appendix A), but not in liver (*p* = 0.07).

Age, length, axial girth, blubber thickness and liver and muscle δ^13^C and δ^15^N were determined for 13 adult male belugas harvested by Inuit partners from the EBS population, aged 19 to 63 years (Table 1). No biological variables were correlated with each other (0.06 ≤ *p* ≤ 0.99), except for δ^15^N liver ratios, which were positively correlated with δ^15^N in muscle tissue (*r_s_* = 0.76; *p* < 0.01). Age, length, axial girth and blubber thickness did not differ between 2009 and 2017 belugas (*p* > 0.05). However, 2009 belugas had lower ratios of muscle δ^13^C (U = 0; *p* < 0.01) and higher ratios of muscle and liver δ^15^N (U = 36; *p* < 0.01) than 2017 belugas. Compared to male adult ringed seals, belugas had lower δ^13^C and higher δ^15^N values in both muscle and liver (0 ≤ U ≤ 117; *p* < 0.01).

### 2.2. Metabolites

#### 2.2.1. Metabolite Profiles

Results of the multiple-factor analysis (MFA) performed with log-transformed metabolite profiles showed that, in both species, each tissue had a distinct metabolite profile except for the two blubber layers (Figure 1 and Figure 2). Indeed, the contribution of metabolite classes did not differ between inner and outer blubber in both ringed seals (0.05 ≤ *p* ≤ 1.00) and belugas (0.05 ≤ *p* ≤ 0.95). Quantitative variables for ringed seals (Figure 1a) and belugas (Figure 1c) indicated that inner and outer blubber were mainly characterized by phosphatidylcholines, fatty acids and sphingomyelins. For liver and plasma, the metabolite classes that distinguished them the most from blubber were amino acids, carbohydrates, biogenic amines and lysophosphatidylcholines.

The average percent contribution of metabolite classes per tissue in both species (Figure 2) indicated that the proportion of lipid classes (i.e., sum of acylcarnitines, fatty acids, lysophosphatidylcholines, phosphatidylcholines and sphingomyelins) in inner and outer blubber in both species was greater than in liver and plasma (all *p* ≤ 0.02), except for the percent contribution of fatty acids that did not differ between liver and inner and outer blubber (*p* ≥ 0.39) in belugas. The sum of lipid classes represented more than 50% of total metabolites in inner (72% in ringed seals and 54% in belugas) and outer blubber (64% in ringed seals and 51% in belugas), while that proportion was lower in liver (29% in ringed seals and 26% in belugas) and plasma (33% in ringed seals and 22% in belugas). Several metabolite classes also differed between plasma and liver profiles. Ringed seal plasma had higher percentages of fatty acids, lysophospatidylcholines and carbohydrates, and lower percentages of amino acids, energy metabolites, bile acids and acylcarnitines compared to liver (all *p* < 0.01). In belugas, the contribution of amino acids, lysophospatidylcholines, phospatidylcholines and sphingomyelins were higher in plasma, whereas contributions of bile acids, energy metabolites and fatty acids were higher in liver (0.02 ≤ *p* < 0.01). The percent contribution of each metabolite class in plasma of belugas did not differ between belugas sampled in 2009 and 2017 (0.14 ≤ *p* ≤ 0.91), except for energy metabolites (U = 0; *p* = 0.04), which were lower, and fatty acids (U = 18; *p* = 0.04) and sphingolipids (U = 18; *p* = 0.04), which were higher in 2009 belugas compared to 2017.

Percent contributions of metabolite classes mostly differed between ringed seal and beluga tissues. All plasma and liver metabolite profiles differed between the two species (0 ≤ U ≤ 99, 0.03 ≤ *p* < 0.01), except for the percent contribution of bile acids (*p* = 0.23) and phosphatidylcholines (*p* = 0.11) in plasma, and liver lysophosphatidylcholines (*p* = 0.41). Metabolite class profiles that differed between inner blubber of ringed seals and belugas were energy metabolites, acylcarnitines, phospatidylcholines, sphingomyelins and carbohydrates (5 ≤ U ≤ 68; 0.03 ≤ *p* < 0.01). In outer blubber, most lipids differed between species, namely acylcarnitines, lysophosphatidylcholines, phospatidylcholines and sphingomyelins (0 ≤ U ≤ 72, *p* < 0.01). Finally, the proportion of the sum of all lipid classes was higher (U = 17; *p* = 0.04) in the inner blubber of ringed seals (72 ± 6.5%) compared to belugas (54 ± 23%), but no differences were observed between the two species in the outer blubber (*p* = 0.09; 64 ± 4.5% in ringed seals and 51 ± 19% in belugas).

#### 2.2.2. Metabolite Concentrations

In ringed seals, the tissue with the highest number of metabolites detected and the greatest concentrations of total metabolites was liver, followed by plasma and then inner and outer blubber (Appendix A). Except for fatty acids (*p* = 0.18), concentrations of each metabolite class were higher in liver compared to the other tissues (14.9 ≤ H(3) ≤ 30.3, *p* < 0.01). Inner and outer blubber had an absence of bile acids, and the metabolite class concentrations did not differ between those two tissues (0.09 ≤ *p* ≤ 0.93). Plasma and liver were mainly characterized by elevated concentrations of carbohydrates (1186 ± 249 and 2552 ± 1615 µg/g, respectively) and energy metabolites (934 ± 344 and 4024 ± 1030 µg/g, respectively). In contrast, inner blubber was characterized by phosphatidylcholines (526 ± 139 µg/g) and fatty acids (308 ± 562 µg/g), while phosphatidylcholines (582 ± 52.4 µg/g) and sphingolipids (226 ± 51.9 µg/g) dominated the outer blubber.

The six metabolites with the highest concentrations were plasma (Σhexose > lactic acid > sphingomyelin C24:1 > glutamine > sphingomyelin C16:0 > phosphatidylcholine acyl-alkyl C38:0); liver (Σhexose > oxaloacetic acid > lactic acid > taurine > sphingomyelin C24:1 > glutamic acid); inner blubber (fatty acid C22:6 > fatty acid C20:5 > lactic acid > taurine > fatty acid C16:1 > oxaloacetic acid); and outer blubber (lactic acid > taurine > sphingomyelin C16:0 > Σhexose > sphingomyelin C24:1 > fatty acid C22:6).

Similar to ringed seals, in belugas, liver had the highest number of metabolites, and metabolite class concentrations were greatest compared to the other tissues (11.5 ≤ H(3) ≤ 28.9, *p* < 0.01) (Appendix A). Inner and outer blubber had also an absence of bile acids, and their metabolite class concentrations did not differ from each other (0.22 ≤ *p* ≤ 1.00). Liver and plasma were mainly characterized by elevated concentrations of carbohydrates (13,345 ± 1607 and 1100 ± 616 µg/g, respectively), followed by energy metabolites (4515 ± 355 µg/g) for liver and amino acids (632 ± 123 µg/g) for plasma. Both inner and outer blubber were mainly characterized by elevated concentrations of phosphatidylcholines (562 ± 259 and 595 ± 309 µg/g, respectively) and energy metabolites (468 ± 898 and 416 ± 764 µg/g, respectively). Plasma concentrations for each metabolite class did not differ between belugas sampled in 2009 and 2017 (0.07 ≤ *p* ≤ 1.00), except for fatty acid (U = 18; *p* = 0.04) and sphingolipid (U = 18; *p* = 0.04) concentrations, which were higher in 2009 than 2017.

In each tissue, the six metabolites with the highest concentrations in belugas were for plasma (Σhexose > glutamine > oxaloacetic acid > alanine > sphingomyelin C24:1 > phosphatidylcholine acyl-alkyl C38:0); liver (Σhexose > lactic acid > taurine > fatty acid C18:0 > oxaloacetic acid > fatty acid C16:1); inner blubber (lactic acid > Σhexose > fatty acid C16:1 > taurine > sphingomyelin C24:1 > alanine; and outer blubber (Σhexose > lactic acid > taurine > fatty acid C16:1 > sphingomyelin C24:1 > alanine).

### 2.3. Contaminants

A total of 124 PCB congeners, 24 PBDE congeners and the emerging flame retardant hexabromobenzene (HBB) were detected in the blubber of the nine male adult ringed seals (Table 2). Σ_124_PCBs contributed to approximatively 86% of the sum of all organohalogen contaminants analyzed, and average concentrations were sixfold higher than for Σ_24_PBDEs. The concentrations of the six most abundant PCB congeners in ringed seal blubber were: CB-168/153 > -138/163/129 > -180/193 > -83/99 > -118 > -113/90/101. The concentrations of the six most abundant PBDE congeners were: BDE-47 > -99 > -100 > -153 > -154 > -28/33. In ringed seals, blubber Σ_124_PCB (*r_s_* = 0.88, *p* < 0.01) and Σ_24_PBDE (*r_s_* = 0.94, *p* < 0.01) concentrations were positively correlated with age (Appendix A) and Σ_24_PBDE concentrations were negatively correlated with muscle δ^15^N (*r^2^* = −0.68, *p* = 0.04) and liver δ^13^C (*r^2^* = −0.70, *p* = 0.04). Total Hg concentrations in ringed seal muscle were positively correlated with girth (*r_s_* = 0.88, *p* < 0.01), blubber thickness (*r^2^* = 0.87, *p* < 0.01) and length (*r^2^* = 0.70, *p* = 0.04). No other correlations were observed between contaminant concentrations and any other biological variable (0.15 ≤ *p* ≤ 0.75).

A total of 191 PCB and 44 PBDE congeners were detected in belugas (Table 2). Σ_191_PCB concentrations contributed to more than 98% of the sum of all organohalogens analyzed, and average concentrations were 100-fold higher than those of Σ_44_PBDE. The concentrations of the six most abundant PCB congeners in beluga blubber were: CB-153 > -138/163 > -101 > -99 > -52 > -118. The concentrations of the six most abundant PBDE congeners in beluga blubber were: BDE-209 > -47 > -100 > -99 > -49 > -154. Total Hg muscle concentrations correlated positively with Σ_191_PCB blubber concentrations (*r_s_* = 0.68; *p* = 0.01) and muscle δ^13^C (*r_s_* = 0.62; *p* = 0.03). Σ_44_PBDE blubber concentrations correlated positively with liver (*r^2^* = 0.77; *p* < 0.01) and muscle δ^15^N (*r_s_* = 0.88; *p* < 0.01). PCB, PBDE and Hg concentrations were not correlated with age of belugas (0.18 ≤ *p* ≤ 0.38). Σ_191_PCB and Hg concentrations did not differ between 2009 and 2017 belugas (*p* = 0.82 and 0.10, respectively); however, belugas in 2009 had greater PBDE concentrations than those in 2017 (U = 36; *p* < 0.01). Total Hg muscle and ΣPCB blubber concentrations did not differ between belugas and ringed seals (*p* = 0.14 and 0.06, respectively), while ΣPBDE concentrations were greater (10-fold) in ringed seals compared to belugas (U = 0; *p* < 0.01).

### 2.4. Correlations between Biological and Contaminant Variables and Metabolite Percent Contributions

Principal component analyses (PCAs) (see Section 4.5) were used as a dimension reduction tool to study the influence of biological variables (age, length, girth, blubber thickness, δ^13^C and δ^15^N) and contaminants (ΣPCB, ΣPBDE, HBB and total Hg concentrations) on the metabolite profiles in plasma, liver, inner blubber and outer blubber of ringed seals and belugas. A total of eight PCAs were performed (i.e., one per tissue and per species) using the log-transformed percent contribution of metabolites quantified in each tissue. The graphic representation of the score and loading plots for each PCA can be found in Appendix A (Appendix A). In ringed seals, a total of six negative correlations were found between explanatory variables and score values (*t*): *t1*_plasma_ correlated with muscle δ^13^C ratios (Figure 3a), *t1*_inner blubber_ (Figure 3b) and *t2*_outer blubber_ (Figure 3c) with log-transformed concentrations of HBB, *t3*_liver_ (14.5%) with girth (*r^2^* = −0.68; *p* = 0.05) and log-transformed total Hg concentrations (*r^2^* = −0.81; *p* < 0.01), and *t3*_outer blubber_ (11.4%) with age (*r_s_* = −0.67; *p* = 0.05).

In belugas, a total of eight significant correlations were found between explanatory variables and PCA scores. Negative correlations were found between *t1*_plasma_ and liver δ^13^C ratios (Figure 3d), *t1*_outer blubber_ with log-transformed total Hg concentrations (Figure 3e), *t2*_outer blubber_ with log-transformed PBDE concentrations (Figure 3f), *t3*_plasma_ (9.84%) with body girth (*r^2^* = −0.69; *p* = 0.02), and *t3*_inner blubber_ (9.13%) with liver δ^13^C ratios (*r^2^* = −0.72; *p* = 0.03). Conversely, *t3*_inner blubber_ was positively correlated with body length (*r^2^* = 0.74; *p* = 0.02) and log-transformed concentrations of total Hg (*r^2^* = 0.83; *p* < 0.01), and for *t3*_plasma_ with log-transformed concentrations of PBDEs (*r^2^* = 0.75; *p* < 0.01). For liver, correlations between explanatory variables and PCA scores were not investigated due to the low number of belugas (i.e., *n* = 4) analyzed for this tissue.

Several of the explanatory variables (i.e., biological and contaminant data) that were correlated with score values (Figure 3) also correlated significantly after a FDR adjustment with the percent contribution of individual metabolites that were associated with the corresponding principal components. In ringed seals, eight phosphatidylcholines in plasma were negatively correlated with muscle δ^13^C ratios, phosphatidylcholine acyl-alkyl C36:5 and sphingomyelin C22:3 in liver were negatively associated with girth, and glutamine, 19 phosphatidylcholines and 6 sphingomyelins in inner blubber were negatively correlated with HBB blubber concentrations (Table 3).

In belugas, acylcarnitine C9 in plasma was associated with girth; two acylcarnitines, two phosphatidylcholines and the sphingomyelin C24:1 in inner blubber were correlated with total Hg muscle concentrations; and asparagine in outer blubber was negatively correlated with PBDE blubber concentrations (Table 4). Several other significant correlations were also found between explanatory variables and metabolites associated with principal components of interest. However, these associations did not pass the FDR adjustment, and could not be considered significant based on this criterion. These correlations can nonetheless be found in the Appendix A (Appendix A), as they may help to improve our understanding of the function of metabolites and the variation of modalities of their profiles in ringed seals and belugas, or in other marine mammal species.

## 3. Discussion

The metabolite profiles in ringed seals and belugas were tissue specific. Inner and outer blubber were characterized by a greater proportion of lipid classes, such as phosphatidylcholines, fatty acids and sphingomyelins, whereas the dominant metabolites in liver and plasma were characterized by high percentages of amino acids, carbohydrates, biogenic amines and lysophosphatidylcholines. Overall, contributions of metabolite classes between ringed seals and belugas were different for plasma and liver, and for inner and outer blubber layers, mostly lipid classes differed (i.e., acylcarnitines, phosphatidylcholines and sphingomyelins), as well as carbohydrates and energy metabolites for inner blubber. Several metabolite profiles in plasma correlated with muscle δ^13^C in ringed seals and girth in belugas, and in both species, metabolite profiles in blubber were correlated with contaminants (HBB for ringed seals, and Hg and PBDEs in belugas).

### 3.1. Inter-Tissue and Inter-Species Metabolite Profiles

Liver in mammals plays a major role in metabolic regulation of dietary nutrients, including fat and carbohydrates [52], while plasma plays a major role in the body distribution of nutrients and metabolites to maintain homeostasis. The high percentages of carbohydrates, energy metabolites and amino acids in liver and plasma tissues in ringed seals and belugas suggest that these tissues are particularly influenced by diet. Similarly, in human plasma, metabolite profiles of amino acids and carbohydrates, as well as other metabolites, including acylcarnitines, lysophosphatidylcholines, phosphatidylcholines and sphingomyelins, were related to diet [53]. Liver in both ringed seals and beluga whales was the tissue with the highest total concentration and number of metabolites detected, with carbohydrates and energy metabolites being dominant. Previous studies have revealed that liver in mammals is rich in a wide range of metabolites, and carbohydrates such as sucrose and lactose are found at very high concentrations [54]. This metabolite richness is consistent with the multiple functions that the liver performs in mammals, such as the synthesis, storage and redistribution of different biomolecules such as lipids, proteins and carbohydrates throughout the body, as well as the detoxification of blood [55]. It is also an important organ for the production of energy.

Carnivorous marine mammals have a protein-based, high-fat and low-carbohydrate diet, and rely primarily on fat for adenosine triphosphate synthesis, transforming dietary fatty acids into glucose via gluconeogenesis [56]. To cover short-term energy needs, marine mammals can also use dietary amino acids as a principal source of energy by producing carbohydrates, but this mechanism is generally dependent on dietary composition (i.e., low-fat prey), percentage of body fat, season, reproductive status or species [56]. In the context of starvation or nutritional deficiency, amino acids can also be used as energy, but are obtained mainly by partial degradation of muscle proteins [57]. Thus, differences in total amino acid percentages observed in plasma and liver of ringed seals and belugas could be explained by differences in nutritional status or diet composition. Ringed seals from this study were harvested just after a fasting period (i.e., during molting), whereas belugas were harvested in middle of summer, corresponding to an active feeding period. Although diet of ringed seals from Lake Melville is not yet well known, it has been suggested that their diet may come from both a marine and brackish food web, consisting of zooplankton and fish species such as shorthorn sculpin (*Myoxocephalus scorpius*), Atlantic cod (*Gadus morhua*), lake whitefish (*Coregonus clupeaformis*) or brook trout (*Salvelinus fontinalis*) [58,59]. Unlike ringed seals from Lake Melville, the diet of EBS belugas seems less diverse, and this population has been reported to feed primarily on marine species, particularly Arctic cod (*Boreogadus saida*) and capelin (*Mallotus villosus*), which have high lipid content [42].

Both blubber layers of ringed seals and beluga whales were dominated by lipid classes (i.e., acylcarnitines, fatty acids, lysophosphatidylcholines, phosphatidylcholines and sphingomyelins), which contributed more than 50% of all metabolites detected in these tissues. Indeed, the blubber of marine mammals is the main storage site for lipids, which serves as an energy reserve, but also has a role in maintaining thermal balance, buoyancy and streamlining [60,61]. Although the vertical lipid composition of marine mammal blubber is known to be heterogeneous, concentrations and contributions of all metabolite classes did not differ between inner and outer blubber layers in both species. In pinnipeds and cetaceans, the outer layer is primarily used for thermoregulatory and structural roles, and thus has a relatively lower lipid content and a higher proportion of collagenous fibers, whereas the inner blubber layer is a highly vascularized, metabolically active layer used for lipid storage and mobilization according to energy demands [60,62]. Although the contribution of fatty acids did not differ between the two blubber layers in both studied species, the proportion of fatty acids in inner blubber was nevertheless higher by at least 10%. Differences in fatty acid composition have also been observed between the blubber layers in ringed seals and beluga whales, with outer blubber composed of a higher proportion of short-chain mono-unsaturated fatty acids with low melting points, and inner blubber being dominated by more saturated fatty acids and long-chain monounsaturated fatty acids (dietary fatty acids) [60,63]. Unfortunately, such an observation could not be made with our dataset due to the limited number of fatty acids assessed in this study. A lipidomic analysis could, however, allow for the determination of a higher variety of fatty acids.

The inner blubber of ringed seals had a higher contribution of total lipid classes (72%) compared to beluga whales (54%). Further, the phospatidylcholines and sphingomyelins had a greater contribution in both inner and outer blubber layers of ringed seals compared to belugas. Since these lipid classes are the main components of cellular membranes [53,64], this could indicate that ringed seal blubber has a higher density of adipocytes compared to beluga whales. Contrary to EBS belugas [65], ringed seals from Lake Melville undergo seasonal fasting during their molting period [66,67], and thus would benefit from having a higher number of adipocytes to store more fat reserve when fasting. The ringed seals from this study were sampled in early May, just after their breeding period and possibly during their fasting period, which would require a mobilization of lipids (lipidogenesis) either to produce energy or, on the contrary, to store it. This could explain the higher percentage of lipid classes in the ringed seal inner and outer blubber compared to belugas.

On an individual basis, the most abundant metabolites found in each tissue of both species were related in some way to energy metabolism. The Σhexose, which represents the sum of all six-carbon monosaccharides with glucose as the major component [53], was among the most abundant metabolites in the plasma, liver and blubber of ringed seals and beluga whales. Glucose is one of the predominant fuels for mammals. Although the main source of glucose is diet, mammals can also temporarily synthetize this fuel from other internal energetic sources such as lipids, lactate or amino acids when food is scarce or during a fasting period [68]. After analyzing blood samples from 21 beluga whales in captivity, Lauderdale et al. [69] proposed that a “normal” blood glucose concentration for a healthy beluga should be between 670 and 1160 µg/g. However, higher plasma glucose concentrations were reported in free-ranging belugas: 1278 ± 616 µg/g for belugas from this study, and 1262 ± 188 µg/g for the Bristol Bay population in Alaska [70]. Additionally, Σhexose concentrations reported in the outer blubber of belugas from the St. Lawrence Estuary, Quebec, Canada (i.e., 3963 ± 182 µg/g) [4] were 17-fold higher compared to values reported in the outer blubber of belugas from this study (i.e., 228 ± 1127 µg/g). This suggests that the determination of a reference value for the glucose concentration in plasma of belugas should be population specific, as it may be influenced by diet or other biological and ecological factors.

Lactic acid, whose conjugate base is lactate, was also one of the most abundant metabolites in ringed seal plasma, and in the liver and blubber layers of both studied species. As the diet of marine mammals is low in carbohydrates [56], this suggests that glucose is synthetized mostly from other substrates such as proteins, lipids and/or lactate. Tartu et al. [46] observed that polar bears from the Barents Sea that had been actively feeding over the winter had greater plasma concentrations of glucose and lactate, suggesting that glucose and lactate levels in ringed seals and beluga whales may be closely related to dietary habits. As the beluga whales that were used in this study were obtained through Indigenous subsistence harvests, the elevated lactate levels observed in their tissues may have also been caused by the increased exertion and stress associated with efforts to escape during the pursuit. In humans, elevated serum lactate concentrations can be the result of mechanisms involved in metabolic acidosis during an intense effort to avoid extensive use of glucose by using lactate as a fuel source for aerobic energy metabolism [71]. An increase in plasma lactate levels has been observed in free-living crocodilians [72], sharpnose sharks (*Rhizoprionodon terraenovae*) [73] or wild dugong (*Dugong dugon*) [74] after capture, and elevated plasma concentrations of lactate dehydrogenase were quantified in spotted dolphins (*Stenella attenuata*) after their first capture [75]. The influence of diet or harvest-related stresses on lactate levels must be interpreted with caution, as baseline lactic acid values are unknown in free-ranging ringed seals and beluga whales.

Taurine was the next most abundant metabolite in liver and inner/outer blubber of ringed seals and beluga whales after Σhexose and/or lactic acid. Taurine is an amino sulfonic acid and is one of the most abundant amino acids in mammals [76]. Although the role of taurine in metabolic function is not known in marine mammals, many physiological roles have been attributed to this metabolite in mammals. For example, it is involved in maintaining the structural integrity of membranes, bile acid conjugation, regulating calcium binding and transport, osmoregulation or as a neuromodulator and as a neurotransmitter [77,78]. Moreover, it is still unclear whether marine mammals can synthesize taurine or whether it is an essential dietary nutrient for these animals [78]. Taurine was also a dominant metabolite in the outer blubber of male St. Lawrence Estuary beluga whales, and concentrations were five times higher (1078 ± 26 µg/g) [4] compared to those from the EBS population (214 ± 137 µg/g), suggesting that taurine levels may vary with factors such as diet. As the St. Estuary beluga population is a heavily contaminated marine mammal population [4,5,79], the differences in taurine and glucose concentrations observed between these two populations may also be related to contaminant exposure.

### 3.2. Contaminants and Effect Thresholds

To date, this is the first time that PCB and PBDE concentrations have been reported in ringed seal tissues from Lake Melville. Average PCB concentrations (1279 ± 1439 ng/g lw) from the present study fell within the range of PCB concentrations reported in male adult ringed seals from Northern Labrador [8], and were three- to sixfold higher than other populations located further north in the Canadian Arctic [25,80]. Similarly, average PBDE concentrations in adult male ringed seals from Lake Melville were approximately 50-fold higher compared to average PBDE concentrations measured in adult male and female ringed seals sampled during 2010 from Saglek Fjord in northern Labrador, and up to 70-fold higher compared to average concentrations reported for other Canadian Arctic populations [25,81]. PCB and PBDE concentrations of ringed seals from Lake Melville were within the same range as other southernmost populations of seals, such as harbor seals (*Phoca vitulina*) from the Gulf of Maine, the Salish Sea and the St. Lawrence Estuary [82,83,84]. For the emerging flame retardant HBB, concentrations from the present study were within the same range as adult male and female ringed seals from Sachs Harbour, NWT (8–30 pg/g lw), Arviat, NU (10–70 pg/g lw), Nain, NL (10–300 pg/g lw), and Resolute Bay, NU (20–110 pg/g lw), but lower than concentrations reported in populations from Arctic Bay, NU (1200–2180 pg/g lw) [81]. Although HBB has been detected in several Arctic air, seawater and sediment samples [85,86], the source, transport and fate of HBB in the Arctic is still relatively unknown.

Muscle Hg concentrations in the adult male ringed seals from Lake Melville (581 ± 488 ng/g ww) were up to fourfold greater than reported concentrations in Northern Labrador adult ringed seals (146–264 ng/g ww), and were among the highest concentrations ever reported since 2010 across the Canadian Arctic (96–690 ng/g ww) [25,87]. These elevated concentrations of PCBs, PBDEs and Hg in the Lake Melville ringed seals are consistent with previous studies showing that, in the Canadian Arctic, southernmost populations tend to have higher concentrations of these biomagnifying contaminants [25,81,87], which likely reflects their closer proximity to urban and industrial areas. This may also explain why ringed seals from Lake Melville had similar PCB and total Hg concentrations and higher PBDEs concentrations compared to EBS belugas, despite them feeding at a lower trophic level.

Overall, PCB and PBDE concentrations in EBS adult male belugas were within the same range as that which was previously reported for this population over the last decade (PCBs: 680–8360 ng/g lw; PBDEs: 5.30–27.3 ng/g lw) [28,30,88]; however, the average total Hg concentrations were lower than those reported previously for adult male belugas from the same population sampled between 2009 and 2012 (1100–1300 ng/g) [30,41].

Total PCB and PBDE concentrations in ringed seals and belugas from the present study were compared with marine mammal effects threshold concentrations (Appendix A). All ringed seals had PCB blubber concentrations above the effects threshold for lymphocyte proliferation in ringed seals (92 ng/g lw) [89]. Three ringed seals (i.e., ringed seals #08, #30 and #31) had PCB blubber concentrations that exceeded several effect thresholds (i.e., 1370 to 2460 ng/g lw) [8], indicating that the transcription of genes involved in xenobiotic elimination, immune response, growth and development (i.e., *Ahr*, *Il1b*, *Igfi* and *Nr3c1*) could be disrupted. For PBDEs, the ringed seal #30 had concentrations exceeding the effects threshold for the suppression of phagocytosis in neutrophils (340 ng/g lw) [89]. PCB blubber concentrations in belugas were all above the PCB effects threshold for phagocytosis in cetaceans (1100 ng/g lw) [89], and all but the two belugas 2017-HI-13 and 2017-HI-14 had concentrations above the effects threshold for disruption of vitamin A and E profiles in belugas (1600 ng/g lw) [29]. PCB and PBDE concentrations in ringed seals and belugas were below the PCB threshold for onset of physiological effects in experimental marine mammal studies (9000 ng/g lw) [90] and for lymphocyte proliferation in belugas (10,200 ng/g lw) [89], as well as the PBDE effects threshold for thyroid hormone disruption in mink (1200 ng/g lw) [91]. To our knowledge, no toxic effects threshold for total Hg in muscle tissue has been determined for marine mammals. Although certain thresholds of effects in the liver and the brain of marine mammals have been proposed, estimating Hg toxicity remains complex and can be difficult to evaluate without other parameters, such as the concentrations of methylmercury and selenium [27,92].

### 3.3. Correlations between Metabolite Profiles, Biological Variables and Contaminants

Metabolite profiles correlated with different degrees with several biological and contaminant parameters in ringed seals and beluga whales across plasma, liver and inner/outer blubber. In ringed seals, several plasma phosphatidylcholine profiles were negatively correlated with muscle δ^13^C. δ^13^C can be used to inform whether individuals are feeding primarily on species of freshwater/coastal species (higher ratios) vs. marine/offshore (lower ratios) origin [93]. Thus, a ringed seal from Lake Melville with a higher proportion of these phospholipids in plasma may be spending more time in estuarine/coastal environments. Profiles of phosphatidylcholine PC ae C36:5 and sphingomyelin C22:3 in ringed seal liver could also be influenced by ecological factors, such as diet, as they were both negatively correlated with girth, which can be indicative of body condition. In polar bears, muscle δ^13^C correlated positively with the liver concentrations of several phosphatidylcholines (i.e., PC aa C34:2, C34:3, C36:2, C36:3, C38:1, C38:3, C40:4, C40:5 and PC ae C36:2, C36:3, C38:2, C38:3), as well as the acylcarnitine C5, indicating that these metabolite classes could be influenced by diet [51]. In belugas from this study, the plasma profiles of nonaylcarnitine (acylcarnitine C9) were positively correlated with the girth of individuals. Moreover, a positive correlation was found a posteriori between plasma profiles of this metabolite and beluga blubber thickness (*r^2^* = 0.79; *p* = 0.006). As girth and blubber thickness can be used as an index of body condition in cetaceans [94,95], these results suggest that plasma levels of nonaylcarnitine could be a potential marker of nutritional status in beluga whales.

HBB concentrations correlated negatively with several phosphatidylcholines and sphingomyelins profiles in the inner blubber of Lake Melville ringed seals. These two phospholipid classes constitute the main components of eukaryotic cell membranes and serve as precursors for signaling molecules in many cellular and physiological processes [64,96]. The blubber of marine mammals is mainly composed of adipocytes, and their size can fluctuate depending on the nutritional status and the body condition of an individual [61]. In summer, Arctic pinnipeds are actively feeding to store high amounts of lipids for winter and/or fasting periods (e.g., breeding, molting). This storage period results in an expansion of adipocyte size and, consequently, more phospholipids are incorporated into the cellular membrane [97]. Moreover, the blubber concentrations of several phosphatidylcholines and sphingomyelins were previously associated with body condition in beluga whales [98]. Thus, the negative correlations observed between HBB concentrations and several phospholipids in ringed seals from Lake Melville could be due to a decrease in body condition. Those correlations could also be the result of its toxicity, inducing lipidosis or a lipid mobilization. After exposing murine 3T3–L1 cell line (adipocytes) and human HepG2 (hepatocytes) cell lines to HBB, Maia et al. [99] reported that this emerging flame retardant induced pronounced effects only in adipocytes, provoking a reduction in adipocyte proliferation, and an increase in lipid accumulation (i.e., increased concentrations of long chain and monounsaturated fatty acids). It is unclear whether these results can be attributed to HBB-related toxicity or the influence of co-factors such as nutritional status or diet. Further research to inform a better understanding of the toxicity of HBB may be warranted, as this xenobiotic was related to several skin gene transcripts involved in endocrine functions and xenobiotic metabolism in the St. Lawrence Estuary beluga population [9].

In EBS belugas, total Hg and PBDE concentrations were correlated with metabolite profiles in inner/outer blubber, respectively. The correlations observed between total Hg and the profiles of acylcarnitines, phosphatidylcholines and sphingomyelin may come from a covariation with common factors related to diet, as beluga Hg concentrations were also correlated positively with muscle δ^13^C ratios. In humans, the plasma profiles of the same lipid classes fluctuate depending on dietary patterns [53]. As Hg only accumulates minimally in the blubber of cetaceans [100,101], and most of the reported toxicological effects of Hg have occurred mainly in other organs such as the brain, kidneys or liver [6], it is unlikely that correlations reported between total Hg muscle concentrations and metabolite profiles in inner blubber of belugas originate from a toxic effect of Hg. Moreover, total Hg concentration is a poor indicator of toxicity in marine mammals [102], as its toxicological effects occur mainly through its methylated form, which has not been determined in this study. Beluga PBDE concentrations were correlated negatively with the profile of asparagine in outer blubber. In obese humans, BDE-47 concentrations in adipose tissue were consistently correlated with plasma metabolites involved in the aspartate and asparagine metabolism [103]. The same PBDE congener exposed to human embryonic kidney cells (HEK293) mainly caused disturbance in energy metabolism by causing a decrease in concentrations of several metabolites, including the ionic form of asparagine, aspartate [104]. Conversely, comparison of the liver metabolite profile in two Canadian Arctic polar bear populations showed that the populations with greater concentrations of PBDEs also add greater concentrations of several amino acids (i.e., alanine, glutamine, leucine, lysine, phenylalanine and valine). More data are required to confirm if PBDEs can disrupt the energy metabolism of belugas.

Despite the use of a conservative statistical approach (i.e., dimensionality reduction and FDR correction), the correlations presented in that section, as well as their interpretations, should be taken with care, as those results may have been impacted by the low sample size used for both species. Moreover, the correlations presented should not be interpreted as cause-and-effect relationships.

### 3.4. Tissue Selection Guidance for Marine Mammal Health Monitoring

When relationships among metabolite profiles and biological/chemical variables prior to the FDR correction in both species are considered (Table 3, Table 4, Appendix A), plasma, liver, and inner blubber emerge as more suitable for assessing the role of diet, whereas inner and outer blubber and liver appear more suitable for studying the role of contaminants. Similar to our observations, several plasma metabolite profiles (i.e., acylcarnitines, amino acids, hexose, phosphatidylcholines, lysophosphatidylcholines and sphingomyelins) were related to diet in humans [53]. The inner blubber of marine mammals has often been a tissue used to study their dietary habits due to its rich composition in long-chain polyunsaturated fatty acids of dietary origin, and its higher metabolic activity compared to outer blubber [61,105,106].

In ringed seals, the liver profile of several metabolites correlated with total Hg concentrations and girth, suggesting that liver seems like a tissue that can be used to study the impacts of ecological changes and contaminant exposure. Indeed, mammal liver is involved in several key metabolic pathways, such as regulation of dietary nutrients, carbohydrate and lipid metabolisms, blood protein synthesis, bile production, detoxification and regulation of inflammation [55]. Due to their composition rich in lipids and their central role in lipid storage and/or metabolism of lipids, liver and blubber are known to accumulate large quantities of lipophilic contaminants such as POPs, brominated flame retardants and mercury.

In beluga whales, similar levels of PCBs and PBDEs were reported in blubber and liver [107]; however, mercury accumulated mainly in liver and minimally in blubber [100]. Ideally, the concentrations of the biomarkers of interest and the contaminants should be obtained in the same tissue. In ringed seals, for example, the proportion of MeHg differed between muscle (14 to 30%) and liver (82 to 97%) due to the ability of marine mammal liver to detoxify Hg in a less toxic form [108,109].

Another important parameter to consider is that due to their different roles and their distinct character, plasma, liver and blubber are likely responding to changes in habitat or exposure to contaminants in different timeframes. In humans, recent diet is a stronger determinant of plasma metabolite profiles [53]. Raach et al. [107] suggested that in beluga whales, liver is a better indicator of recent exposure to organic contaminants, whereas full blubber reflect historical accumulations. Differences in cell renewal rates between plasma, liver or blubber may also be an important parameter to consider, depending on the time scale of the external stressor studied. For example, in humans, hepatocytes renew every 6 to 12 months and adipocytes every 8 years, whereas the renewal of blood cells is only a few days [110].

## 4. Materials and Methods

### 4.1. Field Sampling

Adult male ringed seals (*n* = 10) were harvested in early May 2019 in Lake Melville (NL, Canada) as part of the Inuit subsistence harvest (Figure 4b). A total of 13 adult male belugas from the EBS population were sampled from harvested belugas associated with the community partnered long-term beluga health monitoring program based out of Hendrickson Island, located in the Mackenzie Estuary near the community of Tuktoyaktuk, Northwest Territories (Figure 4a). Samples were obtained in July 2009 (*n* = 4) and July 2017 (*n* = 9) by local Inuvialuit hunters and beluga monitors [111]. Axial and maximum girth, length and blubber thickness were obtained for each species. In collaboration with local hunters, samples of plasma, liver (except for 2017 belugas) and inner and outer blubber (except for 2009 belugas) were collected in the field and flash frozen below −80 °C on site for metabolomics analyses. Muscle (for stable isotope and Hg analyses), liver (for stable isotope analyses) and full blubber (for contaminant analysis) were also sampled, wrapped in aluminum foil and stored at −20 °C on site. The age of ringed seals was determined by Matson’s Laboratory, Manhattan, MT, USA, by counting the annual growth layers in the cementum of a longitudinal thin section of a lower canine tooth using a compound microscope and transmitted light [112]. The age of 2009 belugas was determined from a thin section of a tooth by counting growth layer groups (1 layer = 1 year) in dentine [113], and the age of 2017 belugas was determined by measuring the aspartic acid D:L enantiomer ratio after hydrolysis of the eye lens under acidic conditions [114]. For all samples collected, appropriate permits and community approval were obtained from the Nunatsiavut Government, Nunatsiavut Health and Environment Review Committee and the Department of Fisheries and Oceans Canada.

### 4.2. Metabolomic Analysis

A total of 40 acylcarnitines, 21 amino acids, 13 bile acids, 22 biogenic amines, 16 energy metabolites, 18 fatty acids, 13 lysophosphatidylcholines, 76 phosphatidylcholines, 15 sphingolipids and total hexose were analyzed by SGS AXYS (Sidney, BC, Canada) following a targeted metabolomic approach described elsewhere [115,116] in plasma, inner blubber and outer blubber of ringed seals harvested in 2019 and belugas harvested in 2017 (full list of metabolites analyzed in Appendix A). The same metabolites were also quantified in the liver of ringed seals, and in the plasma and liver of belugas harvested in 2009. Given that the extraction efficiency of metabolites can vary according to the tissue targeted, the extraction efficiency of metabolites has been validated prior to final analyses, according to a validation method described in Benskin et al. [115]. Validation results are available in Benskin et al. [115] for liver, in the internal lab procedure report MLM-4825 for plasma and in the reports MLM-4415 and -8024 for blubber. Metabolites were extracted from tissue samples ground in methanol using a bead blender. Amino acids and biogenic amines were derivatized prior to analysis. All metabolites were quantified using an Agilent 1100 high-performance liquid chromatography (HPLC) system coupled to an API4000 triple quadrupole MS (Applied Biosystems/Sciex, Concord, ON, Canada). Quantification was made by isotope dilution using authentic standards and identical (or homologous) isotopically labeled internal standards, and a quadratic metabolite-specific calibration curve. Specifically, eight calibration samples containing internal and authentic standards were run at the start of MS analysis. Ratios of the authentic standard peak areas and the surrogate peak areas were used to create a calibration curve specific to each metabolite. Methods limits of detection (MLODs) and quantification (MLOQs) were based on the calibration samples containing the lowest metabolite concentrations and were specific to both the metabolite and the sample itself.

Quality control and assurance procedures comprised triplicate analyses of procedural method blanks and SRMs (SC7991, SC6280, SC7447 or SC8297). Metabolites that were detected in one or more blank samples at concentrations >33% of the average sample value, that showed more than 30% variation in concentrations between triplicates for the SRMs or that varied by more than 50% of the expected value of SRMs were excluded from further analyses. Analytes that could not be quantified reliably based on the calibration curve from the standards were also excluded from further analyses. Recovery percentages of SRMs in all tissues and for both species combined varied between 55% and 149% (Appendix A).

### 4.3. Chemical Analyses

#### 4.3.1. Organohalogens

For both ringed seals and belugas, the total concentrations of PCBs and PBDEs were determined from full blubber samples (i.e., a vertical section including outer and inner blubber layers). Ringed seal blubber samples were analyzed for 209 PCB and 52 PBDE congeners, as well as two emerging flame retardants (i.e., HBB and pentabromoethylbenzene (PBEB)) by ALS environmental analytical (Burlington, ON, Canada), whereas blubber samples of 2017 belugas were analyzed for 209 PCB and 46 PBDE congeners by SGS AXYS (Sidney, BC, Canada). For both species, PCB congeners were determined using the USEPA method 1668C EPA [117], and the USEPA method 1614A EPA [118] was used for the determination of brominated flame retardants. Briefly, samples were extracted by Soxhlet extraction overnight using dichloromethane. Extracts were cleaned by Gel-permeation chromatography (GPC) and alumina column chromatography. Procedural blanks and extracts were spiked with ^13^C-labeled surrogate standards (up to 29 for PCBs, 14 for PBDEs and HBB) and analyzed using isotope-dilution high-resolution gas chromatography/high-resolution mass spectrometry (HRGC/HRMS) using a SPB-Octyl column for PCBs and a DB-5ms column for PBDEs. The instrument was mass calibrated using perfluorokerosene (PFK).

Blubber samples of belugas sampled in 2009 were analyzed for 206 PCB and 65 PBDE congeners by the Fisheries and Oceans Canada Laboratory of Excellence in Aquatic Chemical Analysis (Institute of Ocean Sciences, Sidney, BC, Canada). Briefly, samples were ground with anhydrous sodium sulphate, and organohalogens were extracted from a glass column using dichloromethane and hexane (1:1 ratio). The extracts were then evaporated to dryness, weighed, resuspended in dichloromethane and hexane (1:1), and analyzed via HRGC/HRMS using a CP-Sil 19 CB column for PCBs and a DB-5HT column for PBDES. Procedural blanks and extracts were spiked with a mixture of internal standards containing 31 ^13^C-labeled PCBs and 11 ^13^C-labeled PBDEs, to enable precise and accurate quantification using an isotope-dilution method. Details of the GC and MS conditions, the criteria used for chemical identification and quantification and the quality practices can be found in the work of Ikonomou et al. [119]. PCB concentration data for 2009 belugas have already been published elsewhere [28].

Quality control and assurance procedures for organohalogen analyses in both species included analysis of procedural method blanks and duplicate blubber samples for each batch of ten to twenty samples. Recoveries of internal standards added to all samples ranged between 22 and 84% for ^13^C-PCBs (30 congeners), ^13^C-PBDEs (14 congeners), and ^13^C-HBB. Percent recoveries of the laboratory control sample ranged between 91 and 122% for the 27 PCB congeners analyzed, between 40 and 127% for the 52 PBDE congeners, 91% for HBB and 120% for pentabromoethylbenzene. The isotope dilution/internal standard method of quantification was used to determine concentrations of target analytes and correct the concentrations based on the percent recovery surrogates. Sample-specific detection limits were determined from an estimated minimum detectable area consisting of 2.5 times the height of the noise in the m/z channel of interest, converted to an area using the area:height ratio of the corresponding labeled surrogate peak. Contaminants that were detected in one or more blank samples at concentrations >33% of the average sample value, that showed more than 30% variation in concentrations between triplicates for the control samples (i.e., SRMs) or that varied by more than 50% of the expected value of SRMs and/or internal standards were excluded from further analyses. All laboratories used in this study participated in the Northern Contaminants Program (NCP)/Arctic Monitoring Assessment Program (AMAP)-14 inter-laboratory study [120]. Lipid content was determined gravimetrically on a fraction of the extract. All organohalogen concentrations are reported in lipid weight (lw).

#### 4.3.2. Total Mercury

Muscle samples of ringed seals and 2017 belugas were analyzed for total Hg using a Direct Mercury Analyzer DMA-80 evo (Milestone Inc., Shelton, CT, USA) system at the Pacific Science Enterprise Centre (West Vancouver, BC, Canada) and at the University of Manitoba (Winnipeg, MB, Canada), respectively. Briefly, 40–60 mg of partially thawed muscle was sub-sampled after slicing away the outer surface and analyzed directly in the DMA. The detection limit of the DMA was 0.005 ng of Hg. Quality control and assurance procedures included the analysis of two blanks, a replicate and three standard reference materials (i.e., NIST-2976, DOLT-5 and DORM-4) for every batch of 10 samples. The percent recoveries for each standard were 101.3% for NIST-2976, 80.2 ± 0.34% for DOLT-5 and 90.4 ± 1.85% for DORM-4.

Muscle samples of 2009 belugas were also analyzed for total Hg at the University of Manitoba (Winnipeg, MB, Canada) using Cold Vapor Atomic Absorption spectroscopy (CVAAS). Approximatively 150 mg of non-oxidized and partially thawed muscle samples were digested using a hydrochloric/nitric acid mixture heated to 90 °C. The detection limit was 0.005 µg/g of Hg. Quality assurance/quality control was accomplished using the three certified reference materials dogfish muscle (DORM-3), dogfish liver (DOLT-3) and lobster hepatopancreas (TORT-2), two blanks and a sample duplicate every run of ten samples. Certified standard reference materials were analyzed in duplicate in every run. Recovery within 10% of the certified values was used as a batch validation for samples. Duplicates of tissue samples were taken every ten samples with an average difference of ~5%. The comparability of results obtained by the two different methods used for 2009 and 2017 belugas has been evaluated and validated as part of the NCP/AMAP quality assurance and quality control interlaboratory comparison study [120].

### 4.4. Stable Isotope Analysis

Liver and muscle samples of ringed seals and belugas were measured for stable carbon and nitrogen isotope ratios. Ringed seal samples were analyzed at the Element and Heavy Isotope Analytical Laboratories (Windsor, ON, Canada) with an Elemental Analyzer-Isotope Ratio Mass Spectrometer (EA-IRMS), and beluga samples were analyzed at the University of Winnipeg Isotope Laboratory (Winnipeg, MB, Canada) using continuous flow ion-ratio mass spectrometry (CF-IRMS). Briefly, muscle and liver samples were freeze-dried for belugas, dried for three days at 60 °C for ringed seals and homogenized. Lipids were removed for the carbon isotope determination using a 2:1 chloroform/methanol extraction and then dried for analysis. Accuracy was obtained through the analysis of laboratory standards used for calibration of results. 

Carbon- and nitrogen-stable isotope abundances are expressed in delta (*δ*) values as the deviation from standards in parts per thousand (‰) using the following equation:(1)$$\delta_{sample}‰=\left[\left(\frac{R_{sample}}{R_{standard}}\right)-1\right]\times 1000$$
where *R* is the ratio of heavy to light isotope (^15^N/^14^N or ^13^C/^12^C) in the sample and standard. The standard used for carbon analysis was Vienna PeeDee Belemnite, and for nitrogen, the standard was IAEA-N-1 (IAEA, Vienna, Austria) for belugas and atmospheric nitrogen for ringed seals. For each batch of 15 samples, analytical precision of ringed seal samples was determined using sample duplicate and four reference materials: Bovine liver (NIST 1577c), Urea IVA 33802174, Tilapia (internal lab standard) and USGS 40. Precision was ±0.19‰ for δ^13^C and ±0.13‰ for δ^15^N.

### 4.5. Statistical Analysis

All statistical analyses were carried out using R version 4.1.1 (R Core Team, 2021, Vienna, Austria). Shapiro–Wilk and Bartlett’s tests were used to test the normality and homogeneity of variances for each variable (i.e., age, stable isotope ratios, body length, girth, contaminant concentrations and metabolite percent contributions to total metabolite concentrations). Variables that deviated from the normal distribution even after log-transformation were analyzed using non-parametric tests. Differences in metabolite class concentrations/percent contributions between tissues were tested using the non-parametric Kruskal–Wallis rank sum test followed by the pairwise Wilcoxon rank sum post hoc test using a false discovery rate (FDR) *p*-value adjustment. Annual differences between 2009 and 2017 belugas and comparison between belugas and ringed seals for contaminant and metabolite concentrations and/or percent contributions were investigated using the Wilcoxon rank sum test.

As the number and the diversity of metabolites varied between tissues (i.e., plasma, liver, inner blubber, and outer blubber) for each species, only the metabolites that were detected in all tissues were used for the comparison of metabolite class percent contributions between tissues. To better characterize metabolite profiles of each tissue, a multiple-factor analysis (MFA) was performed with log-transformed percent contribution of metabolites present in all tissues, using the package *Factoshiny* [121].

Considering the elevated number of variables (≥629 per species) to examine compared to the sample size, and given the elevated number of correlations (>7400 per species) that would have had to be performed to investigate relationships between metabolite percent contributions (response variables) and contaminant and biological data (explanatory variables), an approach using Principal Component Analyses (PCAs) was chosen in order to reduce the probability of false negatives and/or false positives in our results (alpha risk or type I error). PCA allows reduction of the dimensionality of data by estimating loading factors for each of the variables being used [122]. For each tissue of each species, one PCA was performed using percent contributions (log-transformed) of all metabolites quantified. Correlations between explanatory variables and score values (*t*) of the three first dimensions of each PCA were then evaluated using Pearson or Spearman correlation coefficients depending on the normality of variables. When a correlation was significant, the final step consisted of examining correlations between the explanatory variable and the percent contribution of all metabolites that were significantly correlated with the PCA dimension involved. Due to the elevated number of correlations that needed to be performed for that step, a FDR correction was applied to each set of correlation analysis in order to reduce risks of type I errors. The *p*-values were adjusted using the two-stage sharpened method using a FDR significance threshold of 0.05 [123]. Raw *p*-values were considered significant only if they remained significant after FDR adjustment. Adjusted *p*-values were referred to as *q*-values to avoid confusion with raw *p*-values. Before performing MFAs or PCAs, missing values were imputed using the package *missMDA* [124] when necessary. Linear correlations were measured using Pearson or Spearman’s rank correlation coefficient, depending on the normality of variables.

## 5. Conclusions

This is the first study to examine metabolite profiles using a multi-matrix approach in ringed seals and beluga whales. This new information can serve as a valuable reference for future research involving the use of metabolomics in marine mammals. Results suggest that plasma and liver are more suitable for studying changes in diet, whereas liver and blubber are more suitable for studying the potential impact of contaminants. The metabolic functions involved, the metabolic activity or the cell renewal rates should be considered upstream when selecting a suitable matrix, and will depend on the stressors being assessed. Additional studies are needed to evaluate the influence of seasonal and nutritional changes on the metabolome of marine mammals.

This multi-matrix study shed some light on the factors affecting marine mammal health, and may offer guidance for at-risk species studies where samples are limited to skin/blubber biopsies, such as the Southern Resident killer whales and the St. Lawrence Estuary belugas.

## Figures and Tables

**Figure 1 metabolites-12-00813-f001:**
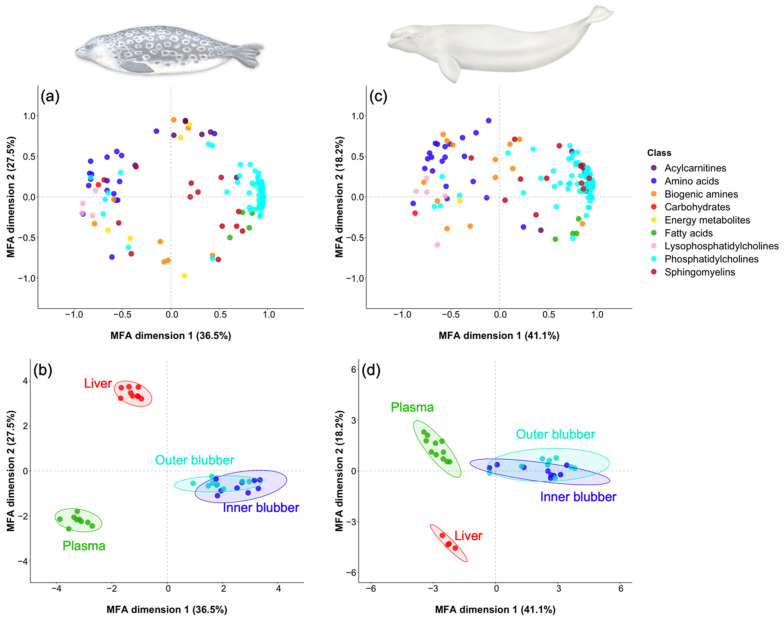
Inter-tissue metabolite patterns differed within each of the two marine mammals. Quantitative variables (**a**,**c**) and individuals (**b**,**d**) of the multiple factor analysis (MFA) was performed with log-transformed percentages of metabolites quantified in plasma, liver, inner blubber and outer blubber of adult male ringed seals from Lake Melville (**a**,**b**) and Eastern Beaufort Sea belugas (**c**,**d**). Quantitative variables and individuals are colored by metabolite classes and tissue, respectively. For each species, only the metabolites that were detected in all tissues were used.

**Figure 2 metabolites-12-00813-f002:**
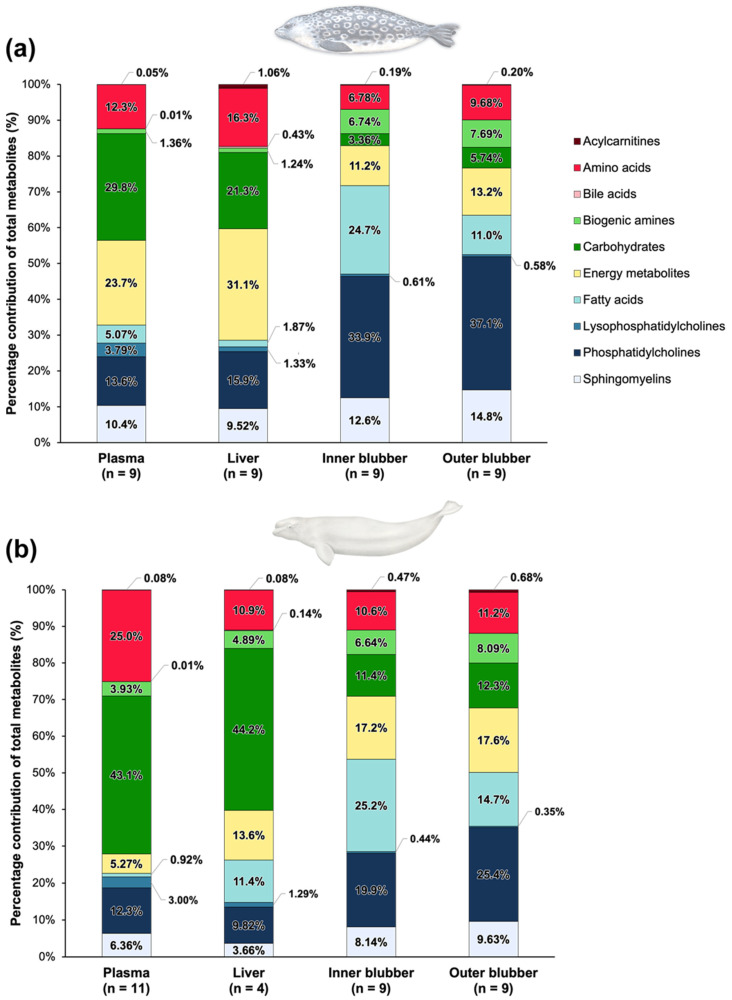
Inner and outer blubber had a greater proportion of lipid classes (i.e., sum of acylcarnitines, fatty acids, lysophosphatidylcholines, phosphatidylcholines and sphingomyelins) relative to plasma and liver. Average percent contribution of metabolite classes in plasma, liver, inner blubber and outer blubber of male adult seals (*n* = 9) from Lake Melville (**a**) and Eastern Beaufort Sea belugas (*n* = 13) (**b**). The plasma metabolite profiles of two belugas sampled in 2009 were considered outliers and therefore were not included.

**Figure 3 metabolites-12-00813-f003:**
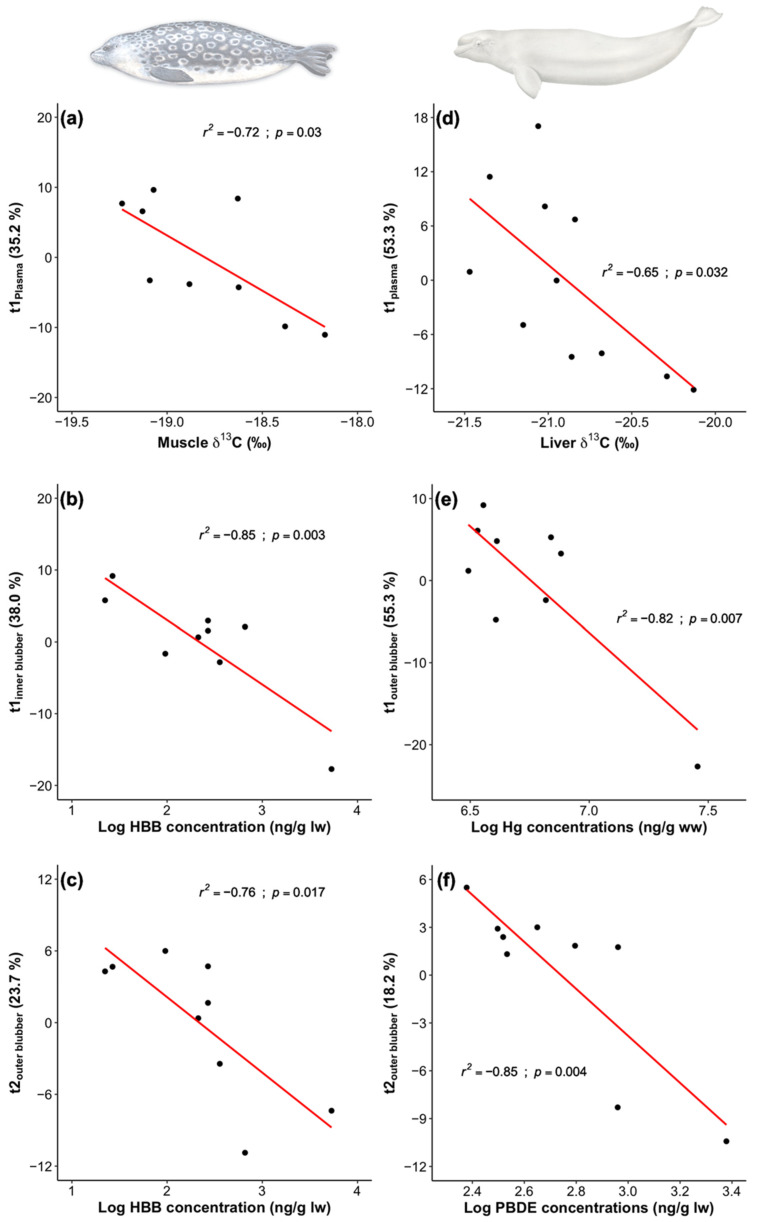
First (*t1*) and second (*t2*) principal components were correlated with muscle δ^13^C (**a**) and HBB concentrations (**b**,**c**) in ringed seals from Lake Melville, and with liver δ^13^C (**d**), total Hg (**e**) and PBDE (**f**) concentrations in Eastern Beaufort Sea belugas. *t1* and *t2* were obtained from the eight PCAs (one per tissue and per species, i.e., plasma, liver, and inner and outer blubber) performed with log-transformed metabolite percent contribution data.

**Figure 4 metabolites-12-00813-f004:**
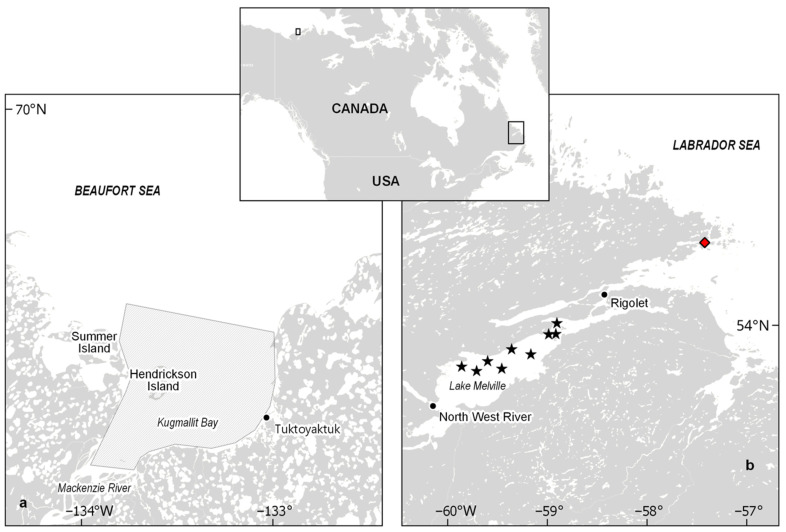
Sampling area (striped polygon) where belugas were harvested in the Beaufort Sea, Canada (**a**), and sampling sites of ringed seals from Lake Melville, Labrador, Canada (**b**). Ringed seals are represented by a black star, and the red diamond represents ringed seal #40, which was considered an outlier (see Section 2.1).

**Table 1 metabolites-12-00813-t001:** Details of individual male adult ringed seals (*n* = 10) sampled from Lake Melville in 2019 and male adult belugas sampled from the Eastern Beaufort Sea population in 2009 (*n* = 4) and 2017 (*n* = 9). Means and standard deviations of ringed seals were calculated without ringed seal #40, which was an outlier.

	Age (Year)	Length (cm)	Axial Girth (cm)	Blubber Thickness (cm)	Muscle δ^13^C (‰)	Muscle δ^15^N (‰)	Liver δ^13^C (‰)	Liver δ^15^N (‰)
*Ringed seals*								
2019-MLV-04	10	152	115	5.50	−19.1	15.5	−18.7	15.2
2019-MLV-08	19	125	100	3.00	−18.6	15.4	−18.7	15.1
2019-MLV-16	6	130	99	3.50	−19.2	15.1	−18.3	15.5
2019-MLV-18	12	137	102	3.50	−18.9	15.5	−19.0	15.0
2019-MLV-28	6	138	106	4.00	−18.4	16.6	−18.4	16.9
2019-MLV-29	6	150	101	5.00	−18.2	15.9	−18.5	15.8
2019-MLV-30	26	132	100	4.00	−19.1	14.7	−18.9	15.2
2019-MLV-31	26	136	100	3.00	−18.6	15.7	−18.7	15.3
2019-MLV-33	7	135	96	3.00	−19.1	15.2	−18.6	15.0
2019-MLV-40	14	244	177	7.00	−17.4	14.8	−17.4	14.8
**Mean ± SD**	**13.1 ± 8.42**	**137 ± 8.79**	**102 ± 5.51**	**3.83 ± 0.90**	**−18.8 ± 0.36**	**15.5 ± 0.54**	**−18.6 ± 0.21**	**15.4 ± 0.60**
*Belugas*								
2009-HI-02	24	432	117	7.60	−20.9	19.1	−20.3	18.3
2009-HI-07	23	366	122	11.4	−23.8	18.7	−20.1	18.3
2009-HI-09	32	422	129	8.90	−23.0	19.2	−21.4	18.1
2009-HI-10	25	427	129	10.2	−21.8	18.8	−20.9	18.2
2017-HI-04	19	437	117	9.20	−18.9	17.5	−21.3	18.0
2017-HI-07	29	417	104	8.90	−18.9	17.2	−20.8	17.6
2017-HI-09	63	442	130	12.1	−19.2	17.3	−21.5	17.6
2017-HI-10	-	433	127	10.8	−19.1	17.3	−20.9	18.0
2017-HI-11	29	406	130	11.4	−19.4	17.3	−20.7	17.8
2017-HI-12	30	420	150	9.50	−19.4	17.2	−21.0	17.6
2017-HI-13	44	424	133	11.4	−19.5	17.0	−20.9	17.9
2017-HI-14	-	399	110	8.30	−19.3	17.2	−21.1	17.6
2017-HI-15	38	419	107	-	−19.3	17.6	−21.1	17.8
**Mean ± SD**	**32.4 ± 12.3**	**419 ± 19.8**	**123 ± 12.5**	**9.97 ± 1.44**	**−20.2 ± 1.65**	**17.8 ± 0.83**	**−20.9 ± 0.39**	**17.9 ± 0.27**

**Table 2 metabolites-12-00813-t002:** Blubber concentrations of ∑polychlorinated biphenyls (PCBs), ∑polybrominated diphenyl ethers (PBDEs), hexabromobenzene (HBB) and muscle concentrations of total mercury (Hg) in adult male ringed seals (*n* = 9) from Lake Melville, and adult male belugas (*n* = 13) from the Eastern Beaufort Sea population.

	ΣPCB ^a,b^ (ng/g lw)	ΣPBDE ^c,d^ (ng/g lw)	HBB ^e^ (pg/g lw)	Total Hg (ng/g ww)
*Ringed seals*				
2019-MLV-04	827	158	3.81	1602
2019-MLV-08	1876	274	11.6	399
2019-MLV-16	884	145	17.0	357
2019-MLV-18	877	247	41.1	487
2019-MLV-28	753	91	10.6	1369
2019-MLV-29	582	80	12.9	931
2019-MLV-30	4587	1082	4.48	636
2019-MLV-31	3586	314	11.4	320
2019-MLV-33	1060	139	7.27	263
**Geometric mean ± SD**	**1279 ± 1439**	**201 ± 311**	**10.5 ± 11.2**	**581 ± 488**
*Belugas*				
2009-HI-02	1769	41.7	n/a	630
2009-HI-07	1777	34.7	n/a	535
2009-HI-09	3465	51.2	n/a	915
2009-HI-10	2447	30.2	n/a	681
2017-HI-04	1907	29.3	n/a	914
2017-HI-07	2253	19.3	n/a	974
2017-HI-09	4964	19.3	n/a	1729
2017-HI-10	2347	12.4	n/a	934
2017-HI-11	2219	14.2	n/a	741
2017-HI-12	1831	12.2	n/a	744
2017-HI-13	1438	12.6	n/a	703
2017-HI-14	1435	10.8	n/a	660
2017-HI-15	2091	16.4	n/a	686
**Geometric mean ± SD**	**2168 ± 955**	**20.6 ± 12.9**	**n/a**	**798 ± 300**

^a^ Sum of 124 PCB congeners for ringed seals: CB-18, -20, -21, -22, -26, -28, -29, -30, -31, -33, -40, -41, -42, -44, -47, -49, -52, -59, -60, -61, -62, -63, -64, -65, -66, -69, -70, -71, -74, -75, -76, -83, -85, -86, -88, -87, -90, -91, -92, -95, -97, -99, -101, -105, -107, -109, -110, -111, -113, -114, -115, -116, -117, -118, -119, -120, -121, -123, -125, -127, -128, -129, -130, -132, -133, -135, -136, -137, -138, -139, -140, -141, -144, -146, -147, -149, -151, -153, -154, -155, -156, -157, -158, -159, -162, -163, -164, -165, -166, -167, -168, -170, -171, -173, -172, -174, -175, -177, -178, -179, -180, -181, -182, -183, -184, -185, -187, -189, -190, -191, -193, -194, -195, -196, -197, -198, -199, -202, -203, -205, -206, -207, -208, and -209. ^b^ Sum of 191 PCB congeners for belugas: CB-1, -2, -4, -5, -6, -7, -8, -9, -10, -11, -12, -14, -15, -16, -17, -18, -19, -20, -21, -22, -23, -24, -25, -26, -27, -28, -29, -30, -31, -32, -33, -34, -36, -37, -38, -39, -40, -41, -42, -43, -44, -45, -46, -47, -48, -49, -50, -51, -52, -53, -54, -56, -57, -59, -60, -61, -62, -63, -64, -65, -66, -67, -68, -71, -70, -72, -74, -75, -76, -77, -79, -81, -82, -83, -84, -85, -86, -87, -88, -89, -90, -91, -92, -94, -95, -96, -97, -98, -99, -100, -101, -102, -103, -104, -105, -107, -108, -109, -110, -111, -112, -113, -114, -115, -118, -119, -120, -121, -123, -124, -125, -126, -127, -128, -129, -130, -131, -132, -133, -134, -135, -136, -137, -138, -139, -140, -141, -143, -144, -145, -146, -147, -148, -149, -150, -151, -152, -153, -154, -155, -156, -157, -158, -159, -162, -163, -164, -165, -166, -167, -169, -170, -171, -172, -173, -174, -175, -176, -177, -178, -179, -180, -181, -182, -183, -184, -185, -186, -187, -188, -189, -190, -191, -192, -193, -194, -195, -196, -197, -198, -199, -200, -201, -202, -203, -204, -205, -206, -207, -208, and -209. ^c^ Sum of 24 PBDE congeners for ringed seals: BDE-8, -11, -15, -17, -25, -28, -33, -35, -37, -47, -49, -51, -66, -77, -99, -100, -118, -119, -120, -126, -140, -153, -154, and -155. ^d^ Sum of 44 PBDE congeners for belugas: BDE-15, -17, -25, -28, -33, -47, -49, -51, -66, -71, -75, -77, -79, -85, -99, -100, -101, -118, -119, -120, -126, -128, -140, -153, -154, -155, -180, -181, -183, -190, -194, -196, -197, -198, -199, -200, -201, -202, -203, -204, -206, -207, -208, and -209. ^e^ HBB was not analyzed in blubber of beluga whales.

**Table 3 metabolites-12-00813-t003:** Correlations between explanatory variables and percent contribution of metabolites quantified in plasma, liver, inner blubber or outer blubber of ringed seals from Lake Melville. Only the correlations that were significant after a FDR adjustment are presented in this table. The full name of each metabolite can be found in Appendix A.

Explanatory Variable	Class	Metabolite	Tissue	Correlation Coefficient	*q*-Value
Muscle δ^13^C	Phosphatidylcholine	PC aa C32:1	Plasma	−0.87	0.033
	Phosphatidylcholine	PC aa C32:2	Plasma	−0.83	0.049
	Phosphatidylcholine	PC aa C34:1	Plasma	−0.86	0.033
	Phosphatidylcholine	PC aa C34:2	Plasma	−0.90	0.033
	Phosphatidylcholine	PC aa C34:3	Plasma	−0.83	0.049
	Phosphatidylcholine	PC aa C36:2	Plasma	−0.88	0.033
	Phosphatidylcholine	PC aa C36:5	Plasma	−0.88	0.033
	Phosphatidylcholine	PC aa C36:6	Plasma	−0.86	0.033
Girth	Phosphatidylcholine	PC ae C36:5	Liver	−0.85	0.027
	Sphingomyelin	SM C22:3	Liver	−0.94	0.003
HBB concentration	Amino acid	Glu	Inner blubber	−0.74	0.045
	Phosphatidylcholine	PC aa C34:1	Inner blubber	−0.94	0.006
	Phosphatidylcholine	PC aa C36:1	Inner blubber	−0.90	0.013
	Phosphatidylcholine	PC aa C36:3	Inner blubber	−0.80	0.029
	Phosphatidylcholine	PC aa C36:4	Inner blubber	−0.80	0.029
	Phosphatidylcholine	PC aa C38:3	Inner blubber	−0.77	0.035
	Phosphatidylcholine	PC aa C38:4	Inner blubber	−0.85	0.017
	Phosphatidylcholine	PC aa C38:5	Inner blubber	−0.85	0.016
	Phosphatidylcholine	PC ae C32:1	Inner blubber	−0.71	0.050
	Phosphatidylcholine	PC ae C34:1	Inner blubber	−0.84	0.018
	Phosphatidylcholine	PC ae C34:2	Inner blubber	−0.78	0.033
	Phosphatidylcholine	PC ae C34:3	Inner blubber	−0.73	0.045
	Phosphatidylcholine	PC ae C36:0	Inner blubber	−0.80	0.029
	Phosphatidylcholine	PC ae C36:1	Inner blubber	−0.77	0.035
	Phosphatidylcholine	PC ae C36:2	Inner blubber	−0.88	0.016
	Phosphatidylcholine	PC ae C36:3	Inner blubber	−0.79	0.029
	Phosphatidylcholine	PC ae C36:4	Inner blubber	−0.85	0.016
	Phosphatidylcholine	PC ae C36:5	Inner blubber	−0.72	0.050
	Phosphatidylcholine	PC ae C38:1	Inner blubber	−0.87	0.016
	Phosphatidylcholine	PC ae C40:2	Inner blubber	-0.93	0.006
	Sphingomyelin	SM (OH) C16:1	Inner blubber	−0.71	0.050
	Sphingomyelin	SM (OH) C22:1	Inner blubber	−0.85	0.016
	Sphingomyelin	SM (OH) C22:2	Inner blubber	−0.75	0.041
	Sphingomyelin	SM (OH) C24:1	Inner blubber	−0.73	0.045
	Sphingomyelin	SM C18:0	Inner blubber	−0.77	0.035
	Sphingomyelin	SM C24:1	Inner blubber	−0.87	0.016

**Table 4 metabolites-12-00813-t004:** Correlations between explanatory variables and percent contribution of metabolites quantified in plasma, liver, inner blubber or outer blubber of Eastern Beaufort Sea belugas. Only the correlations that were significant after a FDR adjustment are presented in this table. The full name of each metabolite can be found in Appendix A.

Explanatory Variable	Class	Metabolite	Tissue	Correlation Coefficient	*q*-Value
Girth	Acylcarnitine	AC C9	Plasma	0.79	0.025
Hg concentration	Acylcarnitine	AC C14:1	Inner blubber	0.82	0.011
	Acylcarnitine	AC C18:2	Inner blubber	0.85	0.010
	Phosphatidylcholine	PC aa C36:0	Inner blubber	0.91	0.004
	Phosphatidylcholine	PC aa C42:1	Inner blubber	−0.76	0.024
	Sphingomyelin	SM C24:1	Inner blubber	0.71	0.032
PBDE concentration	Amino acid	Asn	Outer blubber	−0.89	0.024

## Data Availability

The data presented in this study are available in the article and Appendix A.

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
