# Peer review of "A Multi-Matrix Metabolomic Approach in Ringed Seals and Beluga Whales to Evaluate Contaminant and Climate-Related Stressors"

_metabolites, 2022, doi:10.3390/metabo12090813_

Round 1

Reviewer 2 Report

The authors present an interesting study on several biological parameters, the metabolome and contaminants in different tissues of ringed seals and beluga whales. Overall, the paper reads well and results are presented clearly. Prior to publication there are some points that need to be addressed, mostly related to metabolomics and total Hg measurements, and interpretation of the hence obtained results.  It is also important to stress the drawbacks of the current study and how this may affect results and conclusions.

1.     Introduction

Line 40 - 53: Suggestion to more clearly structure this paragraph by addressing potential adverse health effects first, and exposure second, with mentioning how marine animals are exposed to POPs and Hg through their feed prior to mentioning maternal transfer (= clear storytelling). 

2.     Results

2.1.:  p. 3: Did biological variables demonstrate correlations in the ringed seals? This is mentioned for the belugas, but not the seals.

2.2.: p. 4: When comparing metabolite concentrations between tissues/matrices, it is important to consider potential differences in metabolite recovery in the different matrices. I advise against comparing metabolite levels in different tissues in section 2.2.1 since, based on the M&M section, it appears the used metabolomics methodology was optimized and validated in liver, but not plasma or blubber. Metabolite recovery in plasma and blubber compared to liver is unknown.

3.     Discussion

Overall, it is important to include a section on the drawbacks of the current study, or mentioned whenever wherever relevant: 

o   Sample size (e.g. line 407)

o   4 samples were obtained in 2009 (vs 2017 & 2019). Based on literature, it can be expected that 10 years of storage impedes metabolome stability and reliability of study results, even when stored at -80°C. 

o   Lack of validation of metabolomics in plasma and blubber (?) 

3.1.: please do not compare metabolite levels between tissues, in line with previous comment section 2.2.

Did the authors consider that utilizing blubber reserves (reduction body condition) may lead to redistribution of POPs in the body?

4.     Materials & methods

4.1.: Benskin et al. describe metabolomics in liver tissue, including method validation in liver specifically. Is the method suited and validated for other matrices? Please add the appropriate reference. Or, if this is not available, how can the authors assure the method is also suited for blubber and plasma, which are distinctly different matrices.

4.2.                   

·      Did analysts make use of matrix-specific methods? Were these methods validated? Please add this information.

·      Total mercury in beluga samples was measured in 2 different laboratories. Does this not impede comparability of the hence obtained results?

5.     Conclusions

Here also it is important to stress the drawbacks of the current study and how this may affect results and conclusions.

Reviewer 3 Report

In the presented study, the authors aimed to use metabolomic profiling in different tissues (plasma, liver, and inner and outer blubber) of ringed seals and beluga whales and correlated the profiles with POP contaminants, Hg and δ13C. Metabolite profiles of blubber samples positively correlated with HBB in ringed seals and in belugas with PBDEs and Hg. The authors further conclude that liver and plasma are more suitable tissues for assessing dietary changes while liver and blubber are more suitable for determining effects of contaminants. These results offer guidance for future biomonitoring studies of marine mammals. The introduction provides sufficient details, results are clearly presented and discussion is adequate. Metabolomics studies of marine mammal samples are still scare; therefore, the present study provides novelty in this field and is of interest to the readers of “Metabolites”. Overall, the work is thorough and acceptable for publication.

I only have a few suggestions:

  1. In the results section 2.2, I would first present the metabolites profile and then provide the reader with the metabolite concentrations.
  2. The authors state in the methods that due to low sample size compared to variables, PCA was used to reduce dimensionality and subsequently used FDR correction to reduce risks of type I errors. I understand that it is extremely challenging to obtain larger sample sizes, but in my opinion, the potential effects of low sample size on the correlation results should also be addressed in the discussion.
  3. In the methods for metabolomic analysis (4.2) in order to reproduce this approach, it would be of interest to the reader to include details of the chromatographic method as they are also not stated in reference Benskin et al. [114].

Round 2

Reviewer 1 Report

The authors have answered all my queries and modified the manuscript accordingly.